# A Geometric Method for Improved Uncertainty Estimation in Real-time

Gabriella Chouraqui[*1]   Liron Cohen[1]   Gil Einziger[1]   Liel Leman[*1]

[1]Department of Computer Science, Ben-Gurion University of the Negev, Israel

## Abstract

Machine learning classifiers are probabilistic in nature, and thus inevitably involve uncertainty. Predicting the probability of a specific input to be correct is called uncertainty (or confidence) estimation and is crucial for risk management. Post-hoc model calibrations can improve models' uncertainty estimations without the need for retraining, and without changing the model. Our work puts forward a geometric-based approach for uncertainty estimation. Roughly speaking, we use the geometric distance of the current input from the existing training inputs as a signal for estimating uncertainty and then calibrate that signal (instead of the model's estimation) using standard post-hoc calibration techniques. We show that our method yields better uncertainty estimations than recently proposed approaches by extensively evaluating multiple datasets and models. In addition, we also demonstrate the possibility of performing our approach in near real-time applications. Our code is available at our Github [Leman and Chouraqui, 2022].

## 1 INTRODUCTION

Machine learning models such as neural networks, random forests, and gradient boosted trees are extensively used in domains ranging from computer vision to transportation and are slowly revolutionizing computer science [Niculescu-Mizil and Caruana, 2006, Zhang and Haghani, 2015]. Dealing with uncertainty is a fundamental challenge for all machine learning-based applications. In principle, classifications are always probabilistic, implying that miss-classifications are inevitable.

*Uncertainty calibration* is the process of adapting machine learning models' confidence estimations to be consistent with the actual success probability of the model [Guo et al., 2017a]. The model's *confidence* evaluation on its classifications, i.e., the model's prediction of the success ratio on a specific input, is an essential aspect of mission-critical machine learning applications as it provides a realistic estimate of the classification's success probability and facilitates informed decisions about the *current* situation. Even a very accurate model may run into an unexpected situation, which could then be communicated to the user by the confidence estimation. For example, consider an autonomous driver that uses a model to identify and classify traffic signs. The model is very accurate, and in most cases, its classifications are correct with high confidence. However, one day, it encountered a traffic sign obscured by, e.g., heavy vegetation. In such a case, the model's classification is more likely to be incorrect. Thus, estimating confidence (i.e., uncertainty) is an essential tool for assessing unavoidable risks, enabling system designers to address the risks better, potentially avoiding unexpected and catastrophic implications. Our autonomous driver, for example, may reduce the speed and activate additional sensors until it reaches higher confidence. Thus, indeed, all popular machine learning models have mechanisms for determining confidence that can be calibrated to maximize the quality of confidence estimations [Niculescu-Mizil and Caruana, 2005, Guo et al., 2017b, Kumar et al., 2019] and there is a concentrated effort to calibrate models better and facilitate more dependable applications [Leistner et al., 2009, Sun et al., 2007].

Existing calibration methods can be divided into two types: *post-hoc* methods that preform a transformation that maps from classifiers' raw outputs to their expected probabilities [Kull et al., 2019, Guo et al., 2017a, Gupta and Ramdas, 2021b], and *ad-hoc* methods that adapt the training process to generate better calibrated design [Thulasidasan et al., 2019, Hendrycks et al., 2019a]. Post-hoc calibration methods are easier to apply as they do not change the model and do not require us to retrain a model. That said, ad-hoc meth-

---

[*]Equal Contribution

*Accepted for the 38[th] Conference on Uncertainty in Artificial Intelligence* (UAI 2022).

ods may lead us to better model training in the first place and thus better models. With the success of the two approaches, recent approaches suggest using ensemble methods whose estimation is a (weighted) average of multiple calibration methods [Ashukha et al., 2020, Ma et al., 2021].

This paper presents a post-hoc uncertainty estimation method, but of a different kind. While current post-hoc methods use the model itself as their signal for calibration, we use the training dataset (without modifying the model). More precisely, our method is based on geometric notions calculated on the training dataset. In fact, our geometric choice of the signal is orthogonal to the preformed calibration method in the sense that we can employ it using various post-hoc calibration methods.

Roughly speaking, we examine the geometric distance of the current input from the existing training inputs and use it to estimate the model's confidence. Intuitively, the confidence is high when the current input is close to training set inputs in the same classification and is far from training set inputs with other classifications. Dually, the confidence would be low when there are very close training set inputs with different classifications. To maximize this geometric signal the inputs should be normalized. That is, the size, format, etc. of the images should be consistent. Thus, in this paper, we employ such well behaved datasets.

Our work demonstrates that geometry can facilitate better uncertainty estimations for diverse models and datasets. We first provide an algorithm for calculating the maximal geometric *separation* of an input. However, calculating the geometric separation for an input requires evaluating the whole space of training inputs, making it a computationally expensive method that is not always feasible. For example, an autonomous driver needs to reach decisions within a short time frame to be effective. Therefore, we also suggest a lightweight approximation called *fast-separation*, and show that it provides an approximation of geometric separation.

Our next challenge is to move from a separation value to a confidence estimation. For this, we apply numerical analysis tools. Thus, to obtain a confidence estimation in real-time we only need to apply a regression function to the calculated separation value. Interestingly, our extensive simulation across different models and datasets shows that our geometric-based method yields better confidence estimations when compared to popular libraries used in the industry [Pedregosa et al., 2011] , as well as recently proposed calibration methods [Kumar et al., 2019, Gupta and Ramdas, 2021a, Guo et al., 2017a, Zhang et al., 2020]. Furthermore, our evaluation shows that using our method with the fast-separation approximation allows for multiple confidence estimations per second, making it real-time applicable.

## 2 RELATED WORKS

The dependability of machine learning models is a key challenge in the research community [Johnson, 2018]. Various works demonstrate vulnerabilities in popular machine learning models [Biggio et al., 2014b,a], or show explicit methods to generate adversarial inputs to such models [Zhou et al., 2012]. Unfortunately, such vulnerabilities are fundamental to the field and cannot be avoided.

As mentioned above, uncertainty calibration is about estimating the model's success probability of classifying a given example. Post-hoc calibration methods apply some transformation to the model's confidence (without changing the model) such transformations include Temperature Scaling (TS) [Guo et al., 2017a, Kull et al., 2019], Ensemble Temperature Scaling (ETS) [Zhang et al., 2020], and cubic spline [Gupta and Ramdas, 2021a]. In brief, these methods are limited by the best learnable mapping between the model's confidence estimations, and the actual confidence. That is, post-hoc calibration methods are limited in mapping each confidence value to another calibrated value. In comparison, our method uses geometric distance as a signal for calibration and its improvement over post-hoc calibration is because geometric distances better differentiate than the model's predicted probabilities in the models and datasets included in our evaluation. Another work that uses a geometric distance in this context is [Dalitz, 2009]. There, the confidence score is computed directly from the geometric distance, while we first fit a function on a subset of the data in order to learn the specific behavior of the dataset and model. Moreover, the work in [Dalitz, 2009] only applies to the k-nearest neighbor model, while our method is applicable to all models.

The recently proposed work of [Kumar et al., 2019] uses a fitting function on the confidence values and then divides the inputs into bins of equal size and outputs the function's average in each bin. The work of [Gupta and Ramdas, 2021a] uses a similar idea but divide the inputs into uniform-mass (rather than equal size) bins. It is interesting to note that while most post-hoc calibration methods are model agnostic, recent methods have begun to look on a neural network non-probabilistic output called logits(before applying softmax) [Guo et al., 2017b, Z.Ding et al., 2020, J.Wenger et al., 2019]. Thus, some of the new post-hoc calibration methods are applicable only to neural networks.

Ensemble methods are similar to post-hoc calibration methods as they do not change the model, but they consider multiple signals to determine the model's confidence [Ashukha et al., 2020, Ma et al., 2021]. In principle, ensemble methods complement our approach. For example, one can include our estimator in an ensemble, e.g., by averaging its prediction with other methods. Ad-hoc calibration is about training models in new manners aimed to yield better uncertainty estimations. Important techniques in this category

include mixup training [Thulasidasan et al., 2019], pre-training [Hendrycks et al., 2019a], label-smoothing [Müller et al., 2019], data augmentation [Ashukha et al., 2020], self-supervised learning [Hendrycks et al., 2019b], Bayesian approximation (MC-dropout) [Gal and Ghahramani, 2016, Gal et al., 2017], Deep Ensemble (DE) [Lakshminarayanan et al., 2017], Snapshot Ensemble [Huang et al., 2017a], Fast Geometric Ensembling (FGE) [Garipov et al., 2018], and SWA-Gaussian (SWAG) [Maddox et al., 2019].

Ad-hoc calibration is perhaps the best approach in public as it tackles the core of model's calibration directly. However, because it offers specific training methods it is of less use to large and already trained models, and the impact of each work is limited to a specific model type (e.g., DNNs in [Garipov et al., 2018]). In compression, ad-hoc and ensemble methods (and our own method) often work for numerous models.

Our geometric method is largely inspired by the approach of robustness proving in machine learning models. In this field, formal methods are used to prove that specific inputs are robust to small adversarial perturbations. That is, we formally prove that all images in a certain geometric radius around a specific train-set image receive the same classification [Narodytska et al., 2018, Katz et al., 2017, Huang et al., 2017b, Gehr et al., 2018, Ehlers, 2017, Einziger et al., 2019]. These works are not applicable to uncertainty calibration as they can only produce proves in an offline manner, and thus only to training set inputs rather than to the current input. However, the underlying intuition is that inputs that are geometrically similar should be classified the same also appears in our approach. Indeed, our work shows that geometric properties of the inputs can help us quantify the uncertainty in certain inputs, and that in general inputs that are less geometrically separated and are 'on the edge' between multiple classifications are more error prune than points that are highly separated from other classes. Thus our work reinforces the intuition behind applying formal methods to prove robustness and support the intuition that more robust training models would be more dependable.

# 3 GEOMETRIC CONFIDENCE EVALUATION

This section lays the foundations for a geometric estimation of the model's confidence level on a given instance. Our work assumes that the inputs are normalized. That is, they are fixed-sized images and within the same format. Under such conditions, we can measure the geometric distance between various inputs.

Formally, a model receives a data input, $x$, and outputs the pair $\langle \mathcal{C}(x), conf(x) \rangle$, where $\mathcal{C}(x)$ is the model's classification of $x$ and $conf(x)$ reflects the probability that the classification is correct. Our current work evaluates $conf(x)$ from

a geometric point of view. We estimate the environment around $x$ where points are closer to inputs of certain classifications over the others. In Section 3.1 we define a geometric separation measure, and provide an algorithm to calculate it. We explain that such a computation is too cumbersome for real-time systems, and so we suggest a lightweight approximation in Section 3.2. Finally, Section 3.3 explains how we use the geometric signal to derive $conf(x)$. That is, mapping a real number corresponding to the geometric separation to a number in $[0, 1]$ corresponding to the confidence ratio.

## 3.1 SEPARATION MEASURE

We look at the displacement of $x$ compared to nearby data inputs within the training set. Intuitively, when $x$ is close to other inputs in $\mathcal{C}(x)$ (i.e., inputs with the same classification as $x$) and is far from inputs with other classifications, then the model is correct with a high probability, implying that $conf(x)$ should be high. On the other hand, when there are training inputs with a different classification close to $x$, we estimate that $\mathcal{C}(x)$ is more likely to be incorrect.

Below we provide definitions that allow us to formalize this intuitive account. In what follows, we consider a model $\mathcal{M}$ to consist of a machine learning model (e.g., a gradient boosted tree or a neural network), along with a labeled train set, $Tr$, used to generate the model. We use an implicit notion of distance, and denote by $d(x, y)$ the distance between inputs $x$ and $y$, and by $D(x, A)$ the distance between the input $x$ and the set $A$ (i.e., the minimal distance between $x$ and the inputs in $A$).

**Definition 1** (Safe and Dangerous inputs). *Let $\mathcal{M}$ be a model. For an input $x$ in the sample space we define:*

$$F_{\mathcal{M}}(x) := \{x' \in Tr : \mathcal{C}(x') = \mathcal{C}(x)\}.$$

*We denote by $\overline{F}_{\mathcal{M}}(x)$ the set $Tr \setminus F_{\mathcal{M}}(x)$.*
*An input $x \in \mathcal{X}$ is labeled as* safe *if it is closer to $F_{\mathcal{M}}(x)$ than to $\overline{F}_{\mathcal{M}}(x)$, and it is labeled as* dangerous *otherwise.*

**Definition 2** (Zones). *Let $x$ be a safe (dangerous) point. A* zone *for $x$, denoted $z_x$, is such that for any input $y$, if $d(x, y) < z_x$, then $D(y, F_{\mathcal{M}}(x)) < D(y, \overline{F}_{\mathcal{M}}(x))$ $(D(y, F_{\mathcal{M}}(x)) \geq D(y, \overline{F}_{\mathcal{M}}(x)))$. For each $x$ we denote the maximal such zone by $\mathcal{Z}(x)$.*

In other words, a zone of a safe (dangerous) input $x$, $\mathcal{Z}(x)$, is a radius around $x$ such that all inputs in this ball are closer to an input in $F_{\mathcal{M}}(x)$ $(\overline{F}_{\mathcal{M}}(x))$ than to any input in $\overline{F}_{\mathcal{M}}(x)$ $(F_{\mathcal{M}}(x))$, respectively.

**Definition 3** (Separation). *The separation of a data input $x$ with respect to the model $\mathcal{M}$ is $\mathcal{Z}(x)$ when $x$ is a safe input, and $-1 \cdot \mathcal{Z}(x)$ when $x$ is a dangerous input.*

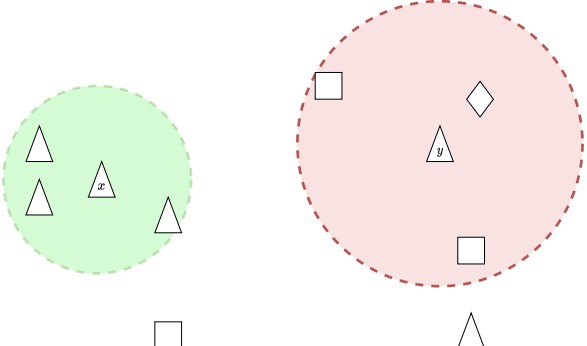

Figure 1: Geometric representation of safe and dangerous inputs, maximal zones, and separation values. The various classifications are illustrated via different shapes.

That is, the separation of $x$ encapsulates the maximal zone for $x$ (provided by the absolute value) together with an indication of whether the point is safe or dangerous (provided by the sign). The separation of $x$ depends only on the classification of $x$ by the model and the train set. This is because our definition partitions the inputs in $Tr$ into two sets: one with $\mathcal{C}(x)$, $F_{\mathcal{M}}(x)$, and one with all other classifications, $\overline{F}_{\mathcal{M}}(x)$. These sets vary between models only when they disagree on the classification of $x$. Note that $x$'s for which the distance from $F_{\mathcal{M}}(x)$ equals the distance from $\overline{F}_{\mathcal{M}}(x)$ are considered dangerous inputs, and their separation measure will be zero.

As mentioned, Definition 2 and Definition 3 use an implicit notion of distance. Such notion can accept any distance metric (e.g., $L_1$, $L_2$ or $L_\infty$). However, in this work, we fix the metric to $L_2$ as it is a standard measure for safety features in adversarial machine learning [Moosavi-Dezfooli et al., 2017], in addition to it being easy to illustrate and intuitive to understand. Furthermore, our methodology relies on calculating the nearest neighbors of a given input, and for $L_2$, this can be done using standard and well-optimized libraries. Accordingly, all our definitions and calculations assume the $L_2$ metrics (Euclidean distances).

Figure 1 provides a geometric illustration of safe and danger zones, and separation values. For illustration purposes, the figure uses the $L_2$ norm with two dimensions, whereas our data usually includes many more dimensions. For example, a $30 \times 30$ traffic sign image will have 900 dimensions. In the figure, $x$ is a safe input, and the green highlighted ball represents its maximal zone which reflects how far we can get from $x$ and still be closer to training set inputs classified the same as $x$ than any other inputs. The input $y$ is a dangerous input, and the red highlighted ball represents its maximal zone which dually represents how far we need to distance ourselves from $x$ so that inputs classified as $x$ become closer than other inputs. Thus, $x$ will have a positive separation value, while $y$ will have a negative one.

Next, we provide a formula for calculating the separation of a given input $x$ within the $L_2$ distance metric.

**Definition 4.** *Given a model $\mathcal{M}$ and an input $x$, define:*

$$\overline{\mathcal{S}}^{\mathcal{M}}(x) = \min_{x'' \in \overline{F}_{\mathcal{M}}(x)} \max_{x' \in F_{\mathcal{M}}(x)} \frac{d^2(x, x'') - d^2(x, x')}{2d(x', x'')}$$

**Lemma 1.** *Let $x, x', x'' \in \mathbb{R}^n$ be inputs such that $d(x, x') < d(x, x'')$. The maximal distance $M(x, x', x'')$ for which if $y \in \mathbb{R}^n$ such that $d(x, y) < M(x, x', x'')$, then $d(y, x') < d(y, x'')$ is*

$$\frac{d^2(x, x'') - d^2(x, x')}{2d(x', x'')}.$$

*Proof.* Since any three points in space define a plane we focus on the plane defined by these three points.

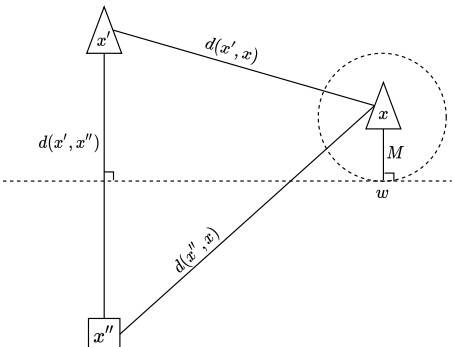

Figure 2: Illustration of the proof of Lemma 1

Figure 2 demonstrates a geometric positioning of the points, and the main constructions in the proof. The perpendicular bisector to the line between $x'$ and $x''$ divides the plane into two parts: one in which all the points are closer to $x''$ than to $x'$ (the lower part in the figure) and one in which all the points are closer to $x'$ than to $x''$ (the upper part in the figure). Our goal is thus to establish the distance between $x$ and the lower part of the plane. Hence, $M(x, x', x'')$ amounts to the distance from $x$ to the perpendicular bisector to the line between $x'$ and $x''$. Using trigonometric calculations, it is straightforward to verify that indeed

$$M(x, x', x'') = \frac{d^2(x, x'') - d^2(x, x')}{2d(x', x'')}.$$

$\square$

**Proposition 1.** $\overline{\mathcal{S}}^{\mathcal{M}}(x)$ *is the separation of $x$ with respect to the model $\mathcal{M}$ (in Definition 3).*

*Proof.* Let $x$ be a safe input, and $y$ be an input such that

$$d(x, y) < \min_{x'' \in \overline{F}_{\mathcal{M}}(x)} \max_{x' \in F_{\mathcal{M}}(x)} \frac{d^2(x, x'') - d^2(x, x')}{2d(x', x'')}.$$

We first show that $y$ is closer to $F_{\mathcal{M}}(x)$ than to $\overline{F}_{\mathcal{M}}(x)$. Let $z'' \in \overline{F}_{\mathcal{M}}(x)$, it suffices to show that there exist $z' \in F_{\mathcal{M}}(x)$ such that $d(y, z') < d(y, z'')$. Notice that

$$d(x,y) < \max_{x' \in F_{\mathcal{M}}(x)} \frac{d^2(x, z'') - d^2(x, x')}{2 d(x', z'')}.$$

Therefore, there exist a $z' \in F_{\mathcal{M}}(x)$ for which

$$d(x,y) < \frac{d^2(x, z'') - d^2(x, z')}{2 d(z', z'')}$$

Thus, since $x$ is a safe point, using Lemma 1, we conclude that $d(y, z') < d(y, z'')$. The proof follows similar arguments for dangerous points, taking the distances as $-\overline{\mathcal{S}}^{\mathcal{M}}$ and flipping the inequalities.

To show maximality, observe that the intersection point marked by $w$ in Figure 2, which is at distance $\overline{\mathcal{S}}^{\mathcal{M}}(x)$ from $x$, can be easily shown to be of equal distances from $F_{\mathcal{M}}(x)$ and $\overline{F}_{\mathcal{M}}(x)$. $\qquad \square$

While separation provides the maximal zone, it is expensive to calculate. As can be seen in Definition 4, to estimate the separation of one specific input, we go over many triplets of inputs. The exact amount is unbounded and depends on the dataset. Thus, separation is infeasible to compute in near real-time. Therefore, when time or computation resources are limited, we require a different and computationally simpler notion. Accordingly, the following section provides an efficient approximation of the separation measure.

### 3.2 FAST-SEPARATION APPROXIMATION

We approximate the separation of a given input using only its distance from $F_{\mathcal{M}}(x)$ and its distance from $\overline{F}_{\mathcal{M}}(x)$. This simplification allows us to calculate a zone for any given point, which is not necessarily the maximal one. The reliance on these two distances enables a faster calculation since we do not perform an exhaustive search over many triplets of inputs. In particular, we do not consider the geometric positioning of the inputs that determine the distance from these sets.

**Definition 5** (Fast-Separation). *Given a model $\mathcal{M}$, the fast-separation of an input $x$, denoted $\underline{\mathcal{S}}^{\mathcal{M}}(x)$, is defined as:*

$$\underline{\mathcal{S}}^{\mathcal{M}}(x) = \frac{D(x, \overline{F}_{\mathcal{M}}(x)) - D(x, F_{\mathcal{M}}(x))}{2}$$

Notice that just as is the case for separation, if $x$ is a safe input, its fast-separation value will be strictly positive and non-positive otherwise.

Figure 3 illustrates the notion of fast-separation. In particular, it exemplifies why it only provides an approximation of the more accurate separation measure. It encapsulates a zone

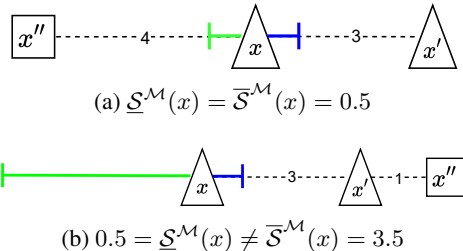

(a) $\underline{\mathcal{S}}^{\mathcal{M}}(x) = \overline{\mathcal{S}}^{\mathcal{M}}(x) = 0.5$

(b) $0.5 = \underline{\mathcal{S}}^{\mathcal{M}}(x) \neq \overline{\mathcal{S}}^{\mathcal{M}}(x) = 3.5$

Figure 3: Geometric representation of the induced zones of $\underline{\mathcal{S}}^{\mathcal{M}}$ and $\overline{\mathcal{S}}^{\mathcal{M}}$ for different input alignments. $\underline{\mathcal{S}}^{\mathcal{M}}$ is represented by blue arrows and $\overline{\mathcal{S}}^{\mathcal{M}}$ by green arrows.

that is less than or equal to that of separation. Sub-figure (a) demonstrates a case in which $\underline{\mathcal{S}}^{\mathcal{M}}(x) = \overline{\mathcal{S}}^{\mathcal{M}}(x)$, while sub-figure (b) presents a case where $\overline{\mathcal{S}}^{\mathcal{M}}(x)$ is considerably larger than $\underline{\mathcal{S}}^{\mathcal{M}}(x)$.

**Proposition 2.** $\underline{\mathcal{S}}^{\mathcal{M}}$ *is a lower bound of the separation* $\overline{\mathcal{S}}^{\mathcal{M}}$ *in the sense that for every safe (dangerous) input* $0 \leq \underline{\mathcal{S}}^{\mathcal{M}}(x) \leq \overline{\mathcal{S}}^{\mathcal{M}}(x)$ $(\overline{\mathcal{S}}^{\mathcal{M}}(x) \leq \underline{\mathcal{S}}^{\mathcal{M}}(x) \leq 0)$.

*Proof.* Let $x$ be a safe input. Since Proposition 1 shows that $\overline{\mathcal{S}}^{\mathcal{M}}(x)$ is the *maximal* zone, it suffices to show that $\underline{\mathcal{S}}^{\mathcal{M}}(x)$ is a zone of $x$. Let $y$ be a point such that

$$d(x,y) < \underline{\mathcal{S}}^{\mathcal{M}} = \frac{D(x, \overline{F}_{\mathcal{M}}(x)) - D(x, F_{\mathcal{M}}(x))}{2}.$$

We show that $D(y, F_{\mathcal{M}}(x)) < D(y, \overline{F}_{\mathcal{M}}(x))$. Take $z' \in F_{\mathcal{M}}(x)$ and $z'', w \in \overline{F}_{\mathcal{M}}(x)$ such that $d(x, z') = D(x, F_{\mathcal{M}}(x))$, $d(x, z'') = D(x, \overline{F}_{\mathcal{M}}(x))$, and $d(y, w) = D(y, \overline{F}_{\mathcal{M}}(x))$. Using the triangle inequality we get:

$$D(y, F_{\mathcal{M}}(x)) \leq d(y, z') \leq d(x, z') + d(x, y)$$
$$< d(x, z') + \frac{d(x, z'') - d(x, z')}{2} = \frac{d(x, z'') + d(x, z')}{2}$$
$$= d(x, z'') - \frac{d(x, z'') - d(x, z')}{2} < d(x, z'') - d(x, y)$$
$$\leq d(x, w) - d(x, y) \leq d(y, w) = D(y, \overline{F}_{\mathcal{M}}(x))$$

For dangerous points, the proof follows similar arguments, switching $F_{\mathcal{M}}(x))$ and $\overline{F}_{\mathcal{M}}(x))$. $\qquad \square$

Proposition 2 shows that the absolute value of $\underline{\mathcal{S}}^{\mathcal{M}}(x)$ is always smaller than or equal to that of $\overline{\mathcal{S}}^{\mathcal{M}}(x)$ and that they have the same sign. Thus, fast-separation is an approximation of separation in the sense that it uses smaller zones. The following proposition further provides an approximation bound for fast-separation.

**Proposition 3.** *The following holds for any point $x$:*

$$|\overline{\mathcal{S}}^{\mathcal{M}}(x) - \underline{\mathcal{S}}^{\mathcal{M}}(x)| \leq \frac{D(x, F_{\mathcal{M}}(x)) + D(x, \overline{F}_{\mathcal{M}}(x))}{2}.$$

*Proof.* We here prove the bound for safe points $x$, the proof for dangerous points is similar. Let $x$ be a safe point. By definition:

$$|\overline{\mathcal{S}}^{\mathcal{M}}(x) - \underline{\mathcal{S}}^{\mathcal{M}}(x)| = \overline{\mathcal{S}}^{\mathcal{M}}(x) - \underline{\mathcal{S}}^{\mathcal{M}}(x) =$$

$$= \min_{x'' \in \overline{F}_{\mathcal{M}}(x)} \max_{x' \in F_{\mathcal{M}}(x)} \frac{d^2(x, x'') - d^2(x, x')}{2d(x', x'')}$$

$$- \frac{D(x, \overline{F}_{\mathcal{M}}(x)) - D(x, F_{\mathcal{M}}(x))}{2}$$

Let $z'' \in \overline{F}_{\mathcal{M}}(x)$ be a point such that $d(x, z'') = D(x, \overline{F}_{\mathcal{M}}(x))$, and let $z' \in F_{\mathcal{M}}(x)$ be a point for which the maximum on the expression above is obtained. Then, we have:

$$|\overline{\mathcal{S}}^{\mathcal{M}}(x) - \underline{\mathcal{S}}^{\mathcal{M}}(x)|$$

$$\leq \max_{x' \in F_{\mathcal{M}}(x)} \frac{d^2(x, z'') - d^2(x, x')}{2d(x', z'')} - \frac{d(x, z'') - D(x, F_{\mathcal{M}}(x))}{2} \tag{1}$$

$$= \frac{d^2(x, z'') - d^2(x, z')}{2d(z', z'')} - \frac{d(x, z'') - D(x, F_{\mathcal{M}}(x))}{2} \tag{2}$$

$$\leq \frac{d(x, z'') + d(x, z')}{2} - \frac{d(x, z'') - D(x, F_{\mathcal{M}}(x))}{2} \tag{3}$$

$$= \frac{d(x, z') + D(x, F_{\mathcal{M}}(x))}{2} \tag{4}$$

$$\leq \frac{D(x, F_{\mathcal{M}}(x)) + D(x, \overline{F}_{\mathcal{M}}(x))}{2} \tag{5}$$

The first inequality (Equation (1)) holds due to the definition of the minimum function. The second inequality (Equation (3)) is due to the triangle inequality. The last inequality (Equation (5)) holds because, since $x$ is a safe point, the maximal distance between $x$ and $z'$ can't be greater than the distance from $x$ to $\overline{F}_{\mathcal{M}}(x)$. $\qquad\square$

Notice that the above bound is tight, in the sense that there exists an example witnessing the exact bound, as shown in Figure 4 below.

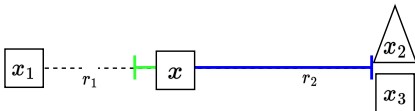

Figure 4: Example of a point $x$ with $|\overline{\mathcal{S}}^{\mathcal{M}}(x) - \underline{\mathcal{S}}^{\mathcal{M}}(x)| = \frac{D(x, \overline{F}_{\mathcal{M}}(x)) + D(x, F_{\mathcal{M}}(x))}{2}$

### 3.3 PREDICTING CONFIDENCE FROM SEPARATION

At this point, we showed how to calculate geometric separation and fast-separation for points (the latter can be done efficiently). Next, we take the calculated (fast-)separation value and derive a confidence estimation *conf*. We use a validation set that is disjoint from the train and test sets to calculate for each input $\overline{\mathcal{S}}^{\mathcal{M}}$ or $\underline{\mathcal{S}}^{\mathcal{M}}$ values. We perform a fitting to map $\overline{\mathcal{S}}^{\mathcal{M}}$ or $\underline{\mathcal{S}}^{\mathcal{M}}$ values to confidence probabilities. The fitting is done between $\overline{\mathcal{S}}^{\mathcal{M}}$ (or $\underline{\mathcal{S}}^{\mathcal{M}}$) values and the ratios of correct classifications (on the validation set) for each unique value. E.g., if for $\underline{\mathcal{S}}^{\mathcal{M}}$ value of 10 we see that 90% of the points are classified correctly then we'll add the pair $\langle 10, 0.9 \rangle$ to the fitting function. We expect very low confidence values for highly negative (fast-)separation values, and we expect to approach 100% confidence when the values become positive enough. The regression function we finally get accepts a (fast-)separation value in $\mathbb{R}$ and outputs a scalar in $[0, 1]$ indicating the confidence estimation, i.e., the predicted success probability for inputs with that (fast-)separation score.

In principle, our method can accept most post-hoc calibration methods to perform the fitting. In this paper, we use isotonic regression as our fitting fuction. Such a function was shown to work best for the tested workloads both for our geometric signal and for the model's original signal, as done in [Pedregosa et al., 2011].

## 4 EXPERIMENTAL RESULTS

This section provides experimental results following the method described in the previous section. We first introduce the datasets, models, and evaluation criteria and then continue to experimental results.

### 4.1 METHODOLOGY

#### 4.1.1 Datasets

Our evaluation uses the following standard datasets.

- *Modified National Institute of Standards and Technology database (MNIST)* [LeCun and Cortes, 2010], which consists of hand-written images designed for training various image processing systems. It includes 70,000 28×28 grayscale images belonging to one of ten labels.

- *Fashion MNIST (Fashion)* [Xiao et al., 2017], which is a dataset comprising of 28×28 grayscale images of 70,000 fashion products from 10 categories.

- *German Traffic Signs Recognition Benchmark (GT-SRB)* [Houben et al., 2013], which is a large image set of traffic signs devised for the single-image, multi-class classification problem. It consists of 50,000 RGB images of traffic signs, belonging to 43 classes.

- *American Sign Language (SignLang)* [Techperson, 2017], which is a database of hand gestures representing a multi-class problem with 24 classes of letters. It consist of 30,000 28×28 grayscale images.

- *Canadian Institute for Advanced Research (CIFAR10)* [A.Krizhevsky et al., 2009], which is a dataset containing 32x32 RGB images of 60,000 objects from 10 classes.

For each dataset[1], we randomly partitioned the data into three subsets: train set $Tr$ (60%), validation set $Vs$ (20%) and test set $Ts$ (20%). The train set is used to calculate fast separation and train the model. The validation set is used to evaluate the confidence estimation associated with each fast-separation value. These values, in turn, are used to fit a Sigmoid function. Finally, the test set is used to evaluate the confidence on new inputs that were *not* present in the train and validation sets.

### 4.1.2 Models

In our evaluation, we use the following popular machine learning models: Random Forest (RF) [Breiman, 2001], Gradient Boosting Decision Trees (GBDT) [Mason et al., 1999], and Convolutional Neural Network (CNN) [Gu et al., 2018]. We chose these models because they are different: RF and GBDT are tree-based, while CNN is a neural network. For RF and GBDT, we configured the meta parameters (e.g., the maximal depth of trees) by cross-validation on the train set. For CNN, we used the configuration suggested by practitioners. Our specific configurations as well as the accuracy scores of each of the models are detailed in Leman and Chouraqui [2022].

### 4.1.3 Evaluation Criteria

To evaluate our method, we compare our (fast-)separation-based confidence estimation to: (a) the built-in confidence in the Sklearn library, (b) the scaling-binning calibrator [Kumar et al., 2019] which we call $SBC$, (c) the histogram binning calibrator [Gupta and Ramdas, 2021a] which we call $HB$, (d) the temperature scaling calibrator [Guo et al., 2017a] which we call $TS$, and (e) the ensemble temperature scaling calibrator [Zhang et al., 2020] which we call $ETS$. $TS$ and $ETS$ are calibration methods for neural networks thus we only apply those to CNNs. Each method received the same baseline model as an input yielding a slightly different calibrated model. Note that our method is evaluated against the uncalibrated model as our method does not affect the model. Moreover, it allows us to compare our method against different calibration methods, as shown in Table 1.

To evaluate the confidence predictions, we use the *Expected Calibration Error (ECE)*, which is a standard method to evaluate confidence calibration of a model [Xing et al., 2020, Krishnan and Tickoo, 2020]. Concretely, the predictions sample of size $n$ are partitioned into $M$ equally spaced

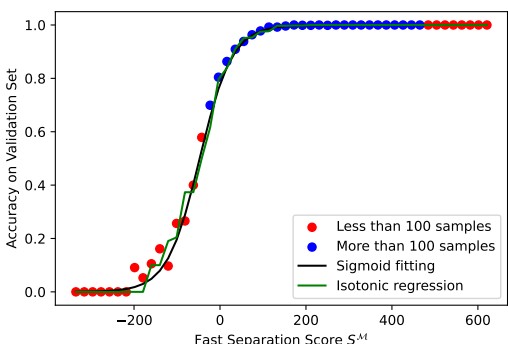

Figure 5: An illustration of the inputs to the fitting function (blue and red dots), and the functions fitted by Sigmoid (black line) and isotonic regression (green line). The inputs are for the MNIST dataset, and the Random Forest model.

bins $(B_m)_{m \leq M}$, and ECE measures the difference between the sample accuracy in the $m^{th}$ bin and the the average confidence in it [Naeini et al., 2015]. Formally, ECE is calculated by the following formula:

$$ECE = \sum_{m=1}^{M} \frac{|B_m|}{n} \left| \mathrm{acc}\left(B_m\right) - \mathrm{conf}\left(B_m\right) \right|$$

where:

- $\mathrm{acc}\left(B_m\right) = \frac{1}{|B_m|} \cdot |\{x \in B_m : \mathcal{C}(x) \text{ is correct}\}|$, and
- $\mathrm{conf}\left(B_m\right) = \frac{1}{|B_m|} \sum_{x \in B_m} conf(x)$.

## 4.2 FITTING FUNCTION

Post-hoc calibration methods based on fitting functions typically use either a logistic (Sigmoid) or an isotonic regression [Zadrozny and Elkan, 2002]. Isotonic regression fits a non-decreasing free-form line to a sequence of observations. In comparison, Sigmoid is a continuous step function. We used both fitting functions on our (fast-)separation values and obtained similar accuracy. We opt here to present the isotonic regression as it provides the best results, as motivated by Figure 5.

Figure 5 illustrates an example of the success ratio of the Random Forest model for MNIST inputs with varying values of $\underline{\mathcal{S}}^{\mathcal{M}}$ scores (similar behavior were observed for the various models and datasets). We clustered inputs with a similar score together (into 50 bins overall) as each classification is correct or not, and we are looking for the average. The black line represents the Sigmoid function and the green line represents the isotonic regression. As can be observed, both regressions are nearly identical on all the points with positive $\underline{\mathcal{S}}^{\mathcal{M}}$ values. We eventually chose isotonic regression because it better fitted the few points with negative $\underline{\mathcal{S}}^{\mathcal{M}}$

---

[1]As is standard practice, we used normalized datasets (e.g., same image size). See our github for details.

Table 1: ECE(%) measures with 95% confidence intervals when varying the calibration method, model and dataset. The parentheses value show the percentage of relative improvement of $\mathcal{S}^{\mathcal{M}}$ over other calibration method.

| Dataset | Model | $\mathcal{S}^{\mathcal{M}}$ | $\overline{\mathcal{S}}^{\mathcal{M}}$ | Sklearn-Iso | Sklearn-Platt | SBC | HB | TS | ETS |
|---|---|---|---|---|---|---|---|---|---|
| **MNIST** | CNN | $.19_{\pm.04}$ | $.19_{\pm.04}$ | - | - | $8.71_{\pm.83(97.8\%)}$ | $.49_{\pm.09(61.2\%)}$ | $.29_{\pm.06(34.4\%)}$ | $.27_{\pm.05(29.6\%)}$ |
| | RF | $.39_{\pm.06}$ | $.40_{\pm.06}$ | $.90_{\pm.12(56.6\%)}$ | $1.48_{\pm.07(73.6\%)}$ | $3.66_{\pm.38(89.3\%)}$ | $.53_{\pm.04(26.4\%)}$ | - | - |
| | GB | $.36_{\pm.07}$ | $.35_{\pm.09}$ | $1.74_{\pm.15(79.3\%)}$ | $1.94_{\pm.13(81.4\%)}$ | $8.23_{\pm.24(95.6\%)}$ | $.48_{\pm.08(24.9\%)}$ | - | - |
| **GTSRB** | CNN | $.40_{\pm.10}$ | $.38_{\pm.07}$ | - | - | $28.44_{\pm2.08(98.5\%)}$ | $.88_{\pm.32(54.5\%)}$ | $1.11_{\pm.40(63.9\%)}$ | $.99_{\pm.41(59.5\%)}$ |
| | RF | $.37_{\pm.04}$ | $.36_{\pm.07}$ | $2.57_{\pm.13(85.6\%)}$ | $4.27_{\pm.14(91.3\%)}$ | $13.71_{\pm.38(97.3\%)}$ | $.81_{\pm.16(54.3\%)}$ | - | - |
| | GB | $.65_{\pm.11}$ | $.67_{\pm.13}$ | $9.96_{\pm.30(93.4\%)}$ | $20.25_{\pm2.17(96.7\%)}$ | $31.08_{\pm.43(97.9\%)}$ | $1.36_{\pm.24(52.2\%)}$ | - | - |
| **SignLang** | CNN | $.01_{\pm.01}$ | $.01_{\pm.01}$ | - | - | $17.83_{\pm.90(99.9\%)}$ | $.22_{\pm.12(95.4\%)}$ | $.25_{\pm.09(96.0\%)}$ | $.24_{\pm.09(95.8\%)}$ |
| | RF | $.08_{\pm.02}$ | $.09_{\pm.03}$ | $.39_{\pm.06(79.4\%)}$ | $1.74_{\pm.08(95.4\%)}$ | $16.88_{\pm.66(99.5\%)}$ | $.19_{\pm.06(57.8\%)}$ | - | - |
| | GB | $.08_{\pm.03}$ | $.08_{\pm.02}$ | $4.05_{\pm0.18(98.0\%)}$ | $5.96_{\pm.17(98.6\%)}$ | $30.97_{\pm.17(99.7\%)}$ | $.47_{\pm.04(82.9\%)}$ | - | - |
| **Fashion** | CNN | $.79_{\pm.13}$ | $.76_{\pm.13}$ | - | - | $7.33_{\pm.51(89.2\%)}$ | $1.93_{\pm.20(59.0\%)}$ | $.84_{\pm.11(5.9\%)}$ | $.88_{\pm.15(10.2\%)}$ |
| | RF | $.74_{\pm.16}$ | $.79_{\pm.10}$ | $.91_{\pm.11(18.6\%)}$ | $3.74_{\pm.12(80.2\%)}$ | $3.45_{\pm.31(78.5\%)}$ | $1.08_{\pm.15(31.4\%)}$ | - | - |
| | GB | $.73_{\pm.13}$ | $.73_{\pm.08}$ | $3.80_{\pm.20(80.7\%)}$ | $5.71_{\pm3.91(87.2\%)}$ | $3.90_{\pm.46(81.2\%)}$ | $1.06_{\pm.14(31.1\%)}$ | - | - |
| **CIFAR-10** | CNN | $1.27_{\pm.19}$ | $1.20_{\pm.15}$ | - | - | $3.57_{\pm.40(64.4\%)}$ | $5.99_{\pm.26(78.7\%)}$ | $5.16_{\pm.22(75.3\%)}$ | $5.16_{\pm.43(75.3\%)}$ |
| | RF | $1.15_{\pm.24}$ | $1.19_{\pm.23}$ | $3.25_{\pm.28(64.6\%)}$ | $4.59_{\pm.24(74.9\%)}$ | $2.99_{\pm.26(61.5\%)}$ | $2.51_{\pm.39(54.1\%)}$ | - | - |
| | GB | $1.25_{\pm.21}$ | $1.31_{\pm.16}$ | $7.57_{\pm.25(83.4\%)}$ | $8.39_{\pm.18(85.1\%)}$ | $2.70_{\pm.34(53.7\%)}$ | $2.80_{\pm.24(55.3\%)}$ | - | - |

values. Interestingly, these points were consistently a poor fit for the Sigmoid regression rendering slightly less accurate on average. Also, observe that the transition is around the value 0, indicating that the distinction of safe and dangerous points is meaningful in confidence evaluation.

## 4.3 CONFIDENCE EVALUATION

Table 1 presents the main experimental results of our work. The table summarizes ECEs for our method (with bin size 30).

Each entry in the table describes the ECE, the 95% confidence interval, and (in parenthesis) the improvement of our fast-separation-based method over each competitor method. The improvement is calculated using the ratio between the difference between our ECE and the competitor's ECE. In this experiment, we perform ten random splits of the data into train, validation, and test sets for each model and dataset. We then measure the ECE of the confidence estimation for all test set items, average the result and take the 95% confidence intervals.

First, observe that $\mathcal{S}^{\mathcal{M}}$ and $\overline{\mathcal{S}}^{\mathcal{M}}$ yield very similar ECEs, and that the differences between them are usually statistically insignificant. Thus, we conclude that $\mathcal{S}^{\mathcal{M}}$ is a very good approximation of $\overline{\mathcal{S}}^{\mathcal{M}}$ despite being considerably simpler to compute. The next interesting comparison is between $\mathcal{S}^{\mathcal{M}}$ and SKlearn. We use the same fitting function (Isotonic regression) in both cases, but SKlearn performs the calibration on the model's natural uncertainty estimation, and $\mathcal{S}^{\mathcal{M}}$ performs the calibration on geometric distances. Thus, the benefit of our approach stems from the geometric signal and not from the chosen fitting function.

Observe that our $\mathcal{S}^{\mathcal{M}}$ improves the confidence estimations consistently and across the board when compared to SKlearn, SBC, and HB. Specifically, we derive improvements up to 99% in all tested models, and for all tested datasets. Such results demonstrate the potential of geometric signals to improve the effectiveness of uncertainty estimation.

## 4.4 REAL-TIME COMPUTATION

Table 2 provides the computational advantage of our method. We used a Macbook Pro with an Intel Core i5 with four processor cores@2.3 GHz and 8GB RAM in this experiment. We measure the throughput of confidence evaluations in predictions per second and the 95% confidence intervals using five trials for each measurement.

Observe that the dominant factor in operation speed is the dataset. These differences are due to variations in training set sizes, where larger training sets result in slower operation. Importantly, our method runs in 23–46 predictions per second in all but the CIFAR-10 datasets. Such performance is within the ballpark for camera-based applications. For reference, a TV is broadcast in 60 frames per second, and most animation films use up to 24 frames per second. Thus our performance is within an applicable scale. CIFAR-10 is considerably larger, and thus our performance on that dataset is a bit slow. While we can also use parallelism to obtain a faster runtime and mitigate this issue, we plan to address larger training sets in future work.

Table 2: Number of confidence estimations per second for the $\underline{\mathcal{S}}^{\mathcal{M}}$-based method with $95\%$ confidence intervals.

| Dataset | Model | Predictions per second |
|---|---|---|
| **MNIST** | CNN | 22.73 $_{\pm 1.90}$ |
| | RF | 22.42 $_{\pm 0.66}$ |
| | GBDT | 22.91 $_{\pm 0.46}$ |
| **GTSRB** | CNN | 25.29 $_{\pm 0.69}$ |
| | RF | 23.23 $_{\pm 1.19}$ |
| | GBDT | 21.78 $_{\pm 2.18}$ |
| **SignLang** | CNN | 46.61 $_{\pm 0.18}$ |
| | RF | 45.16 $_{\pm 2.71}$ |
| | GBDT | 46.84 $_{\pm 0.40}$ |
| **Fashion** | CNN | 22.99 $_{\pm 0.16}$ |
| | RF | 22.85 $_{\pm 0.03}$ |
| | GBDT | 23.20 $_{\pm 0.31}$ |
| **CIFAR10** | CNN | 7.08 $_{\pm 0.24}$ |
| | RF | 6.76 $_{\pm 0.27}$ |
| | GBDT | 6.84 $_{\pm 0.41}$ |

## 5 CONCLUSION

Our work uses post-hoc calibration techniques but on a geometry-based signal rather than on the model's confidence estimation. We demonstrated the feasibility of our approach in estimating uncertainty for multiple models, and for multiple datasets. Our evaluation shows that our fast-separation method ($\underline{\mathcal{S}}^{\mathcal{M}}$) outperforms post-hoc calibration methods based on the model's confidence consistently and across the board. Our approach reduces the error in confidence estimations by up to 99% compared to alternative methods (depending on the specific dataset and model).

In addition, we showed that for moderately-sized standard datasets our method achieves near-real time operation. As suggested by our analysis and indicated by the experimental results, the complexity of calculating $\underline{\mathcal{S}}^{\mathcal{M}}$ depends on the training-set size which implies that very large datasets would be slower, and not run in near real-time. Another related limitation of the work presented in this paper is that our current approach requires using also the training set. Since shipping the model with as little communication or storage overhead is important, our future research focuses on utilizing a smaller data structure that approximates the entire training set. Thus, we will develop ways to control the effect of the dataset size on the run-time by calculating confidence estimations only on a subset of the dataset. E.g., by pre-processing the data and removing data inputs that are geometrically close to each other and reducing the overheads.

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
