# OpenReview forum: "A Geometric Method for Improved Uncertainty Estimation in Real-time"
_auai.org/UAI/2022/Conference — UAI 2022 Poster_

### Official Review · Reviewer_8Gqc · 2022-04-11

**Q2(1) Originality/Novelty:** 2
**Q2(2) Significance/Impact:** 2
**Q2(3) Correctness/Technical Quality:** 3
**Q2(6) Clarity Of Writing:** 3
**Q6 Overall Score:** 5
**Q8 Confidence In Your Score:** 3

**Q1 Summary And Contributions:**

The authors propose a geometric approach for model re-calibration, based on computing the distance or separation of instance from the training set and then using this information as predictive feature for the accuracy to perform re-calibration using standard post-hoc methods. The authors then discuss the results of an experimental evaluation showing promising results for the proposed calibration method.

**Q2 Assessment Of The Paper:**

More detailed information regarding each of these aspects is given below:

**Q2(4) Quality Of Experiments (Optional):**

3: Good: The experimental evaluation is adequate, and the results convincingly support the main claims.

**Q2(5) Reproducibility:**

3: Good: Key resources (e.g., proofs, code, data) are available and key details (e.g., proofs, experimental setup) are sufficiently well-described for competent researchers to confidently reproduce the main results.

**Q3 Main Strengths:**

- The proposed approach is clearly described and motivated
- Code is publicly available online (or was at some point during the reviewing period)
- The experimental analysis shows promising results of the proposed approach

**Q4 Main Weakness:**

- Theoretical analysis of the proposed approach is limited
- The novelty of the proposed calibration method is limited, as it simply consists of using a different predictive feature to predict accuracies: standard post-hoc calibration models are still needed for correcting the confidence scores
- Statistical analysis of the results is limited

**Q5 Detailed Comments To The Authors:**

The proposed approach seems interesting: even though the novelty is more on the definition of "predictive features" for calibration (indeed, the proposed separation is used as predictive feature for any given post-hoc calibration method), it seems that the use of the proposed geometric information can be helpful in improving calibration, even more so than the base confidence scores predicted by the underlying model.
The theoretical analysis of the fast-separation algorithm is relatively limited: the authors describe this as an approximation for "genuine" separation, but do not study error or approximation bounds for this approach. Maybe it would be better to call fast-separation a "bounding approach" since it only provides a bound for separation without approximation guarantees?
The proposed method seems to be better than the considered state-of-the-art methods on all the considered datasets and for all of the considered models: this is rather interesting as one rarely sees such an overwhelming improvement. Do the authors believe there may be any risks of data leakage or similar forms of overfitting? In any case, tabular format is probably not the most helpful way to visualize the obtained results: maybe it would be clearer and more useful to represent the results in graphical format (e.g., through a critical difference diagram)?
Finally, still in regard to the experimental evaluation: why these specific datasets have been selected? Why these specific comparison methods? The authors should provide greater detail on this aspect to avoid making the choices seem arbitrary.
Also, while previously I managed to access the code supporting the publication at the anonymous Github linked in the paper, now the same link says "The repository is expired".

**Q7 Justification For Your Score:**

The article is interesting and the proposed geometric approach to calibration seems to be useful, nonetheless the paper is in the present stage too limited for publication: in particular, the experimental analysis and the adopted benchmarking choices should be better explained and motivated, and the comparison with state-of-the-art should be expanded.

**Q9 Complying With Reviewing Instructions:**

1: Yes.

---

### Official Review · Reviewer_o892 · 2022-04-12

**Q2(1) Originality/Novelty:** 3
**Q2(2) Significance/Impact:** 3
**Q2(3) Correctness/Technical Quality:** 3
**Q2(6) Clarity Of Writing:** 4
**Q6 Overall Score:** 5
**Q8 Confidence In Your Score:** 3

**Q1 Summary And Contributions:**

 This paper proposes a post-hoc uncertainty calibration method for classifiers based on a geometric signal derived from the train data set and the output of the classifier.


**Q2 Assessment Of The Paper:**

More detailed information regarding each of these aspects is given below:

**Q2(4) Quality Of Experiments (Optional):**

3: Good: The experimental evaluation is adequate, and the results convincingly support the main claims.

**Q2(5) Reproducibility:**

4: Excellent: Key resources (e.g., proofs, code, data) are available and key details (e.g., proof sketches, experimental setup) are comprehensively described for competent researchers to confidently and easily reproduce the main results.

**Q3 Main Strengths:**

Although the use of geometrical properties of the data points to compute the confidence of the classifiers (as well as accept/reject criterion) is not completely new, the measure proposed in this paper and the way it is used it to calculate the confidence of the classifier is novel to the best of my knowledge.

The concept is clearly explained, the statements are easy to follow through, the paper is well-written in general.

The problem considered here is an interesting and important.

**Q4 Main Weakness:**

Although the authors claim and show by experiments that their proposition works better than the confidence computation in act in sklearn as well as SBC and HB, there is little motivation on why they have chosen this way of computing the confidences.

Another main weakness in my opinion is the fact that this approach seems to work on certain datasets which are "normalized" as the authors describe. I would like to know if there is a possible way to normalize a dataset and then use this approach, and how it would affect the outcome.

**Q5 Detailed Comments To The Authors:**

- Is there a way to use this approach on a dataset which is not normalized, say with images of different sizes?

- [Dalitz 2009] lists a comprehensive list of geometrical "signals" used to compute the confidence for KNN classifiers, which seem to share the fundamental ideas used in this paper. I would like to see how this approach compares to those methods (some of which are pretty old.)

- Is there a classification (a characterization) of the datasets where the fast-separation behaves close enough to the exact approach? Adding such a discussion, or even some theoretical results in this direction would significantly improve your work.

 [Dalitz 2009]: Dalitz, Christoph. "Reject options and confidence measures for knn classifiers." Schriftenreihe des Fachbereichs Elektrotechnik und Informatik Hochschule Niederrhein 8.2009 (2009): 16-38.

**Q7 Justification For Your Score:**

Please see above comments.

**Q9 Complying With Reviewing Instructions:**

1: Yes.

---

### Official Review · Reviewer_Biea · 2022-04-12

**Q2(1) Originality/Novelty:** 2
**Q2(2) Significance/Impact:** 2
**Q2(3) Correctness/Technical Quality:** 3
**Q2(6) Clarity Of Writing:** 2
**Q6 Overall Score:** 5
**Q8 Confidence In Your Score:** 3

**Q1 Summary And Contributions:**

The authors proposed a post-hoc method to estimate uncertainty in machine learning classifiers. Unlike existing post-hoc methods that use raw model outputs like logits as a signal for calibration, the proposed method develops a notion of zone defined by implicit distance measures and uses it as a signal for uncertainty calibration. The authors measure a quantity called "separation" for every point, which indicates whether a point is safe/dangerous and quantifies the maximal safe/dangerous zone


**Q2 Assessment Of The Paper:**

More detailed information regarding each of these aspects is given below:

**Q2(4) Quality Of Experiments (Optional):**

2: Fair: The experimental evaluation is weak: important baselines are missing, or the results do not adequately support the main claims.

**Q2(5) Reproducibility:**

3: Good: Key resources (e.g., proofs, code, data) are available and key details (e.g., proofs, experimental setup) are sufficiently well-described for competent researchers to confidently reproduce the main results.

**Q3 Main Strengths:**

Although uncertainty calibration has been analyzed from different viewpoints in prior work, a geometric approach is interesting.

**Q4 Main Weakness:**

Weak empirical evaluations (see comments below)/analysis and details.

**Q5 Detailed Comments To The Authors:**

Experimental evaluation is weak in the following aspects.
1.	Lack of comparison with standard methods like TS, ETS etc.
2.	Lack of details about the models used: (a) The paper mentions only the types of models but does not provide the source of the models i,e pre-trained or trained from scratch or at least the link to config.txt does not work. (b) The architecture of the models used for comparison on CIFAR 10. (c) Accuracy of the models.
3.	The first method proposed is not scalable, and the fast method, although it performs well for smaller datasets, may not be scalable for larger datasets. Comparison with other standard methods can help better estimate the efficacy of this approach.
4.	From a practical viewpoint, this approach requires the presence of model, training data and their corresponding evaluations to estimate uncertainty in prediction for a single data point. The model cannot be shipped in as a calibrated classifier.
5.     The link describing the models has expired.


**Q7 Justification For Your Score:**

The score stems from the weaknesses mentioned above. Unless this method is thoroughly compared to existing methods, this approach cannot produce a standalone calibrated classifier.

**Q9 Complying With Reviewing Instructions:**

1: Yes.

---

### Official Review · Reviewer_FTBu · 2022-04-14

**Q2(1) Originality/Novelty:** 3
**Q2(2) Significance/Impact:** 3
**Q2(3) Correctness/Technical Quality:** 3
**Q2(6) Clarity Of Writing:** 3
**Q6 Overall Score:** 7
**Q8 Confidence In Your Score:** 2

**Q1 Summary And Contributions:**

The authors propose a geometric-based approach for uncertainty estimation. The distance of the current input from the existing training inputs is used to estimating uncertainty. The further an input is from training inputs, the more uncertain we ought to be about it. The authors then calibrate using standard methods.

**Q2 Assessment Of The Paper:**

More detailed information regarding each of these aspects is given below:

**Q2(4) Quality Of Experiments (Optional):**

3: Good: The experimental evaluation is adequate, and the results convincingly support the main claims.

**Q2(5) Reproducibility:**

3: Good: Key resources (e.g., proofs, code, data) are available and key details (e.g., proofs, experimental setup) are sufficiently well-described for competent researchers to confidently reproduce the main results.

**Q3 Main Strengths:**

Interesting new approach for uncertainty estimation based on geometric distances between test and training points.
Competitive with existing calibration methods, but more efficient. Good set of experiments. Nice writing and visualization of ideas. Implementation is available.

**Q4 Main Weakness:**

There could be more baselines and ways to perform calibration (both in baselines and in the proposed method).

**Q5 Detailed Comments To The Authors:**

The paper is interesting and proposes a new approach. It would be better to embed the work (and contrast it) with existing method classes.

**Q7 Justification For Your Score:**

Interesting new approach to uncertainty estimation.

**Q9 Complying With Reviewing Instructions:**

1: Yes.

---

### Decision · Program_Chairs · 2022-05-15

**Decision:**

Accept (Poster)

**Comment:**

Meta Review: The paper introduces a geometric-based approach for post-hoc uncertainty estimation.  It received four generally positive reviews, with three borderline accepts and one accept.  The reviewers believe the proposed geometric approach is novel and efficient, and the experiments are extensive and demonstrate the competitive performance of the proposed method.  Some concerns were raised, including further evaluation of the method, unclear performance on unnormalized data, and additional theoretical and statistical analysis of the method.